# Accuracy of O-RADS System in Differentiating Between Benign and Malignant Adnexal Masses Assessed via External Validation by Inexperienced Gynecologists

**DOI:** 10.3390/cancers16223820

**Published:** 2024-11-13

**Authors:** Peeradech Buranaworathitikul, Veera Wisanumahimachai, Natthaphon Phoblap, Yosagorn Porngasemsart, Waranya Rugfoong, Nuttha Yotchana, Pakaporn Uthaichalanont, Thunthida Jiampochaman, Chayanid Kunanukulwatana, Atiphoom Thiamkaew, Suchaya Luewan, Charuwan Tantipalakorn, Theera Tongsong

**Affiliations:** Department of Obstetrics and Gynecology, Faculty of Medicine, Chiang Mai University, Chiang Mai 50200, Thailand

**Keywords:** adnexal mass, benign tumor, malignant tumor, O-RADS US system, ultrasound

## Abstract

The O-RADS system demonstrates high diagnostic performance in distinguishing benign from malignant adnexal masses, even when used by inexperienced examiners. However, the false positive rate remains relatively high, mainly due to the over-interpretation of solid-appearing components in classic benign lesions. Despite this, inter-observer variability among non-expert raters was substantial. Incorporating O-RADS system training into residency programs is beneficial for inexperienced practitioners. This study could be an educational model for gynecologic residency training for other systems of sonographic features.

## 1. Introduction

The preoperative differentiation between benign and malignant adnexal masses is widely recognized as essential for clinical decision making, as the management of these conditions requires distinct approaches. For instance, functional ovarian masses typically necessitate no intervention beyond observation, whereas endometriomas and dermoid cysts are generally amenable to laparoscopic surgery. Conversely, the management of malignant masses necessitates the involvement of oncologists or a well-coordinated consultation with a gynecologic oncologist or referral to a tertiary care center with specialized surgical expertise. Numerous ultrasound evaluation systems and protocols have been developed to facilitate the discrimination between benign and malignant adnexal masses. The most frequently utilized systems include the risk of malignancy index (RMI); the international ovarian tumor analysis (IOTA) system and its variants, such as the IOTA simple rules and logistic regression model; the ADNEX model; and the Ovarian-Adnexal Reporting and Data System Ultrasound (O-RADS US) risk stratification.

The RMI was initially developed by Jacobs et al. [1] in 1990 and subsequently modified to RMI-2 by Tingulstad et al. [2] in 1996. In 2008, Timmerman et al. [3] introduced the IOTA simple rules to distinguish between benign and malignant ovarian masses based on sonographic features indicative of malignancy (M-rules) or benignity (B-rules). This framework evolved into diagnostic logistic regression models [4] and the ADNEX model [5], which also incorporates clinical parameters. The IOTA system is predominantly practiced in Europe. The O-RADS US risk stratification system provides a lexicon and classification for ovarian lesions, encompassing six categories (O-RADS 0–5) that range from normal to high risk of malignancy [6]. This system was developed by the ACR Ovarian-Adnexal Reporting and Data System Committee to accurately assess the malignancy risk of ovarian masses. O-RADS US is relatively new and mostly studied in the United States.

To the best of our knowledge, a limited number of published studies have evaluated O-RADS US, and most were retrospective [7,8,9,10] and validated by certified or expert sonographers. The validation in two studies was based on static images and included only surgical cases [7,8]. One study assessed the inter-observer variability in performing O-RADS US by experienced examiners (fellowship/staff members) [10]. Importantly, only a very limited number of studies on O-RADS US involving inexperienced examiners and interpreters have been published. To date, only one study, reported by Zhou et al. [11], evaluated the accuracy of the learning curve of O-RADS US among practitioners with varying levels of experience after a one-week training course. However, this study was based on the review and scoring of static images obtained from retrospective data. Accordingly, external validation of O-RADS US for differentiating benign from malignant adnexal masses using both surgical and non-surgical treatments as the reference standard, particularly among non-experienced practitioners, is very limited in the literature.

The RMI and ADNEX models necessitate clinical data such as CA-125 levels or menopausal status, thereby gaining popularity among gynecologists but less so among sonographers and radiologists. The IOTA simple rules, based solely on sonographic features, appeal to sonographers; however, they have been reported to yield a significant number of inconclusive results (up to 20–30%) [12,13], despite their high sensitivity and specificity. In contrast, O-RADS, which is also based solely on sonographic features and is not associated with inconclusive results, is preferred by both sonographers and gynecologists. Nonetheless, O-RADS is relatively new and has primarily been applied by highly skilled examiners, with limited external validation and testing by inexperienced practitioners. Before O-RADS can be widely adopted by general practitioners, it must be evaluated for its accuracy and reproducibility when used by less experienced clinicians. Accordingly, we conducted this study aimed to assess the diagnostic performance of O-RADS in distinguishing malignant from benign adnexal masses when employed by inexperienced examiners in different populations.

## 2. Materials and Methods

This study was conducted using a prospective database of gynecologic patients undergoing ultrasound examinations for adnexal masses at Maharaj Nakorn Chiang Mai Hospital, a tertiary center within the Department of Obstetrics and Gynecology, Faculty of Medicine, Chiang Mai University, between March 2018 and August 2024. The study was ethically approved by the institutional review board (Research Ethics Committee 4, Faculty of Medicine, Chiang Mai University; study code ID: OBG-OBG-2566-0074, approval date 13 June 2023).

The study included three components: (1) the development of a prospective database by second-year obstetrics and gynecology residents, considered non-expert examiners, which had been created prior to the initiation of this study; (2) training the second-year residents with a four-week O-RADS US course (20 h) at the beginning of their second year of residency; and (3) O-RADS scoring based on video clips retrieved from the database, conducted by the participating residents at the end of their second year of residency.

The database development consisted of two parts: part 1, a prospective database from our IOTA project, established prior to this study; and part 2, a newly developed database created by the authors. Both parts were created under the same protocol, each including two separate files: a clinical database and a companion database of ultrasound DICOM files. All ultrasound examinations in both parts were performed by second-year residents under the supervision of the same two experienced examiners (S.L., T.T.).

During database development, patients were counseled and invited to participate in the study. Each patient provided written informed consent. Patients included in the database were those presenting with adnexal masses and undergoing ultrasound examinations. All examinations were conducted using a transvaginal approach and an additional transabdominal approach as necessary, employing real-time 5–7.5 MHz transvaginal or 2.5–5 MHz transabdominal curvilinear transducers connected to a Voluson E8 or Voluson E10 machine (GE Medical Systems, Zipf, Austria). During the ultrasound examination, the morphology and vascularization of the masses were evaluated using 2D real-time ultrasound and color flow mapping. The results were prospectively recorded in research forms and entered into the database. The ultrasound clips and still images included all the dimensions of the adnexal masses, both 2D real-time and color flow mapping images, and measurements of the three greatest dimensions of the masses. Patients’ demographic data, clinical data, surgical findings, and pathological or intra-operative diagnoses were also included. A companion DICOM file database of ultrasound images and cine loops of the relevant patients was created separately. The final diagnosis, considered the gold standard, was based on pathological examinations of surgical specimens, intra-operative diagnoses made by the surgeons in some cases of benign disorders without pathological specimens, or clinical diagnosis upon follow-up examination by senior gynecologists in cases of non-operative management, such as functional ovarian masses, pelvic abscess treated with medications, etc.

This study comprised two phases: the training phase for the residents participating in the study and the O-RADS US scoring phase. In phase 1, ten residents attended a training course on the O-RADS US lexicon and risk stratification at the beginning of their second year of residency. The O-RADS training included lectures, demonstrations, and interactive tests conducted by an experienced sonographer certified in the IOTA system (TT). The training utilized 50 anonymized ultrasound video clips and 100 images of adnexal masses with known diagnoses for practice in describing O-RADS US features and assigning O-RADS category. The course content also included a detailed explanation of all terms, following the O-RADS US lexicon and classification system, which encompasses all risk categories of adnexal masses as recommended by the Consensus Guideline from the ACR Ovarian-Adnexal Reporting and Data System Committee [6]. In phase 2, at the end of the residents’ second year of residency (the O-RADS rating phase), the records of patients, along with the relevant ultrasound clips that met the inclusion criteria, were retrieved from the databases. These records were reviewed and displayed case by case, by the senior author (TT), to the ten inexperienced examiners (residents) simultaneously for evaluation and O-RADS scoring. The residents, who were blinded to the final diagnoses, were presented with both video clips (cine loops) and ultrasound images of adnexal masses. They evaluated and categorized the masses into five categories (O-RADS 1 to 5) based on sonographic features and recorded their ratings in predefined record forms, which included lexicon descriptors listed on a checklist in an Excel sheet, as shown in Figure 1. The residents only needed to tick the option that best matched the sonographic features (lexicon descriptors) and O-RADS category for each case. Note that the training and evaluation process was limited to second-year residents. After completing a training course at the start of their second year, the ten non-expert raters continued with their routine residency in obstetrics and gynecology. They rotated through various assignments as part of their standard training. Although they did not receive additional specialized training during the second year, they might have encountered gynecologic ultrasound as a component of their regular residency training and participated in O-RADS-US assessment at the end of the second year of residency training. The sets of video clips and images used for their assessment had never been used in the training course.

All adnexal masses were sonographically categorized into two main groups: benign and malignant. Masses with a pathological diagnosis of low-malignant-potential tumors were classified as malignant. Exclusion criteria were as follows: 1. patients with already known diagnoses of adnexal masses; 2. patients undergoing surgery more than one week after the ultrasound examination, except cases with no surgical treatment; 3. uncertainty of final diagnosis; 4. prior treatment before the ultrasound examination; 5. pregnancy.

### Statistical Analysis

Inter-observer variability in sonographic categorization among the ten examiners was assessed for agreement using the multi-rater Fleiss Kappa coefficient. The diagnostic performance of the O-RADS was calculated and presented as a receiver operating characteristic curve (ROC); its sensitivity and specificity were also calculated, along with 95% confidence intervals. The overall accuracy in predicting malignancy was determined by consensus score: the median score for each case from the 10 raters was used to represent the collective evaluation, using O-RADS category 4 as the threshold cut-off for prediction. Statistical analysis was performed using the Statistical Package for the Social Sciences (SPSS) software version 26.0 (IBM Corp. Released 2019. IBM SPSS Statistics for Windows, Armonk, NY, USA: IBM Corp.). A *p*-value of less than 0.05 was considered statistically significant.

## 3. Results

During the study period, a total of 303 women underwent ultrasound examination for adnexal masses. Of these, 201 adnexal masses met the inclusion criteria and were evaluated by the ten residents, as shown in Figure 2. The various final diagnoses of the masses are presented in Table 1. Most of them (136 cases) were classified as benign, accounting for two thirds of the cases, while one third of cases (65 cases) were classified as malignant.

The performance of O-RADS in distinguishing benign from malignant masses for each examiner and as a consensus among them, based on the median O-RADS category rated by the ten residents and using category 4 as a threshold, is presented in Table 2 and Figure 3. Overall, the sensitivity and specificity were 90.8% (95% CI: 82.2–96.2%) and 86.8% (95% CI: 80.4–91.8%), respectively. The performance of each examiner was comparable. The area under the ROC curve did not differ significantly (Z-test, paired samples; all *p*-values > 0.05, as presented in Figure 3, Table 3). The agreement in predicting benign and malignant cases among the ten raters was good, with a Fleiss Kappa coefficient of 0.642, as presented in Table 4. However, the agreement in O-RADS categorization was moderate, with a Fleiss Kappa coefficient of 0.471.

Notably, the false positive rate in predicting malignant masses was relatively high at 13.2%. Over-categorization was mostly confined to cases involving complex endometrioma or dermoid cyst with complex adhesion masses and tubo-ovarian abscesses, as shown in Table 5. Of them, based on subjective assessment by the experienced authors, the false ratings were often due to mistaking solid-appearing amorphous masses for solid tissue, leading to upscoring to a more severe category. Among the cases with false negatives, most were associated with dermoid cysts with an immature component or struma ovarii.

## 4. Discussion

The insights gained from this study are as follows: (1) The diagnostic performance of O-RADS US is very good among non-expert examiners who completed a training course on ultrasound for adnexal masses, including video clips of various sonographic features. (2) There is a good inter-rater agreement among non-expert examiners in assigning O-RADS categories, with a Fleiss Kappa value of 0.642, especially in the assignment of O-RADS 4 and O-RADS 5. (3) The degree of agreement in assigning individual features was fair, with a Fleiss Kappa value of 0.36–0.56. (4) The most common cause of false positives for malignancy prediction is the presence of solid components in classic benign cases, which were mistaken for solid tissue indicative of malignancy. Among these, the classic benign masses, which included infected or complicated endometriomas and dermoid cysts or pelvic abscesses, were relatively common and were complex with markedly thickened walls or septa, simulating solid tissue rather than an amorphous mass or blood clot. Additionally, some vascularization was found in these inflammatory areas. Therefore, this group of adnexal masses was likely mistaken for malignancy, based on the worst-case scenario rule that if there is doubt or uncertainty about whether a mass is solid or amorphous, it should be interpreted as solid tissue, suggesting a higher O-RADS category. (5) Based on the ROC curve, the threshold of O-RADS 4 is appropriate for malignancy prediction, providing a sensitivity of 90.8% and a specificity of 86.8%.

In most studies in the literature concerning the accuracy of various systems in distinguishing benign from malignant masses, the ultrasound examiners and interpreters were experienced sonographers or gynecologists. Indeed, these examiners do not require predictive systems, as subjective assessment is highly effective in experienced hands. However, the diagnostic challenge in clinical practice often involves general practitioners or general gynecologists, rather than gynecologic sonographers or oncologists. The essential knowledge needed is the accuracy of such systems—whether IOTA simple rules, RMI, ADNEX models, or O-RADS US—when applied by less experienced hands. Accordingly, we conducted this study, which differs from most previous studies, with the primary aim of evaluating the accuracy of O-RADS when used by non-experienced doctors, in the hope that this knowledge could be applied to general gynecologists or practitioners worldwide.

Upon reviewing the existing literature, the performance of the O-RADS US system when used by certified sonographers or experienced practitioners varies across studies. Most studies report excellent sensitivity, ranging from 91% to 100% [7,8,9,10,14,15,16,17,18,19,20,21]. However, specificity varies more widely, typically between 80% and 96% [7,8,10,14,15,18,19,20,21], with some studies reporting much lower specificity, from 46% to 75% [9,16,17]. The primary limitation of the O-RADS US system appears to be the inconsistency in specificity, or the false positive rate. This inconsistency may be related to the examiner’s experience level, the quality of the ultrasound equipment, and familiarity with the O-RADS system. In our study, the accuracy of the O-RADS US system was comparable to or slightly lower than that found in most studies, despite the fact that both the ultrasound examinations and the ratings were handled by residents in training rather than expert examiners. Notably, the performance of our non-experts in this study was comparable to that reported by Guo et al. [15], who found that the O-RADS system had a sensitivity of 91.0% and a specificity of 84.8%. In their study, the reviewers were two senior and two junior doctors, whereas our study involved ten raters who were all in a residency training program. These findings underscore the importance of training in the sonographic evaluation of adnexal masses, suggesting that such training can enable non-expert examiners to achieve diagnostic accuracy closer to that of experts.

Based on this study, the main problem with O-RADS ratings among non-experts is a relatively high false positive rate, leading to unnecessary consultations with oncologists or referrals to tertiary centers. The false positive rates were primarily associated with mistaking amorphous solid blood clots or dense fibrotic masses susceptible to inflammation or infection, commonly seen in complicated classic benign masses, for solid tissue. This misinterpretation resulted in higher categorization and increased inter-observer variability. This is likely due to the tendency to assume the worst-case scenario in the face of uncertainty. In other words, most false positives were linked to classic benign masses with complications, such as twisting, fibrotic mass susceptible to inflammation, or intra-mass hemorrhage, which create echogenic areas within the masses and reduce confidence in correctly identifying them as benign. The findings from this study suggest that improving O-RADS requires a greater focus on familiarizing practitioners with benign fibrotic masses, thickened cystic walls or septa, and various forms and stages of blood clots.

Interestingly, inter-observer variability was good for the main categorization of O-RADS and for differentiating between benign and malignant group (Fleiss Kappa coefficient of 0.642), while it was only fair for specific sonographic features and overall O-RADS classification (Fleiss Kappa coefficient of 0.471). Nevertheless, the main objective of O-RADS is to differentiate benign from malignant masses and to assign the main categories. Therefore, such a problem is of less concern. This variability is mostly associated with inconsistency in the rating of benign groups, especially cystic lesions and classic benign masses. For example, in cases of endometrioma with recent hemorrhage, raters might classify it as a classic benign endometrioma, a classic benign hemorrhagic mass, or a unilocular cystic lesion without solid components. The variability observed among less experienced examiners may be attributable to several factors, including difficulties in interpreting deviations from classic imaging features, inconsistencies in training, or the inherent complexity of the O-RADS classification system, particularly within the benign category. Increased exposure to variations in imaging for each type of tumor, including greater familiarity with both classic and atypical features through video demonstrations, along with expert consultations during routine ultrasound assessments of adnexal masses, could potentially enhance rater consistency.

Our results indirectly suggest that exposure to ultrasound clips of adnexal masses under supervision can significantly improve the ability to distinguish between benign and malignant masses, bringing their skills closer to the level of an expert during the training year, instead of requiring many years of practice. There is convincing evidence that extensive exposure to ultrasound clips with proper supervision could effectively shorten the learning curve. Therefore, in modern gynecology residency training, each institute should include a course that provides multiple exposures to ultrasound features of adnexal masses or develop their own database of video clips as a learning tool, with mentors available for supervision. Familiarity with sonographic features and known definitive diagnoses is a key component of successful ultrasound training.

**The strengths** of the study are as follows: (1) Although the O-RADS ratings in this study were not based on real-time examinations, the assessments were made using video clips, ensuring that all raters were exposed to exactly the same material. This approach is ideal for evaluating inter-observer variability, unlike some previous studies that relied on static ultrasound images [8] rather than cine loops, which more closely represent real-time evaluations. (2) The sample size was adequate for assessing both the performance of the tests and inter-observer variability. (3) Both the original examiners and the interpreters were non-experts, meaning our results could reflect the accuracy of the O-RADS US system among non-experts. (4) All interpreters independently rated and were blinded to the final diagnosis of the adnexal masses. (5) As an external validation by a different group of practitioners was used, our results may better reflect the reproducibility of the O-RADS US system.

**The weaknesses** of the study are as follows: (1) The raters did not interpret the O-RADS in real-time, as it would be unethical to conduct multiple ultrasound examinations on the same patient by ten different doctors. However, the impact is likely minimal, as all raters reviewed the same video clips simultaneously for interpretation. (2) The two parts of the database were developed by different teams of gynecologic residents using different ultrasound machine models. While this could theoretically affect the quality of the ultrasound videos, the impact was likely minimal. Both teams had similar levels of experience, were supervised by the same staff members, and collected data prospectively using the same inclusion criteria. (3) We included no comparisons of the performance of O-RADS US between non-expert and expert examiners or between O-RADS US and other diagnostic systems like IOTA or ADNEX models.

**Research implication:** The performance of the O-RADS system may be enhanced or developed into an O-RADS-plus version by incorporating parameters that are missing from the original O-RADS, such as the presence of ascites, bilaterality, and acoustic shadowing. These parameters can be easily assessed during the same examination without additional effort and by both sonographers and clinicians. Future studies should explore such a modified O-RADS system. Additionally, the added value of clinical factors that are commonly available in practice, such as patients’ age, menopausal status, or CA-125 levels, should be further investigated as optional or additional parameters for the O-RADS US system.

**Clinical implication:** This study suggests that the O-RADS US system is simple and can be effectively applied by non-expert examiners, although its performance might not be perfect. Importantly, practicing with O-RADS US seems to aid examiners during their learning curve by helping them recognize certain sonographic features systematically, which may lead to a more rapid development of skills in evaluating adnexal masses compared to simply scanning. In actual practice, O-RADS US is straightforward and can assist clinicians in triaging patients for appropriate management. Additionally, while the O-RADS system relies exclusively on sonographic features to differentiate between benign and malignant masses, clinicians and gynecologists can further enhance decision making by incorporating additional clinical predictive factors. These may include patient age, menopausal status, and tumor markers, particularly CA-125 levels. Integrating these factors can be especially beneficial when sonographic findings are inconclusive or ambiguous. Furthermore, developing a user-friendly application for daily practice or integrating the O-RADS US system into a preset function of ultrasound machines as built-in software could be beneficial.

Finally, this study emphasizes the significance of targeted exposure to sonographic features, as gained through structured training courses, in enhancing the learning curve for less experienced practitioners. Consequently, tailored training programs or streamlined systems that provide extensive exposure to sonographic variations—either through video clips of classic cases from established databases or through real-time cases encountered in daily practice—are likely to substantially improve diagnostic skills.

## 5. Conclusions

The training course on the O-RADS US system, introduced at the beginning of the second year of residency, has shown the potential to significantly enhance the diagnostic skills of non-expert examiners. Within just one year of practice, these skills can reach levels close to those of more experienced practitioners. Moreover, external validation by non-expert examiners indicated that the O-RADS US system achieved a sensitivity of 90% and a specificity of 86%. These skills are likely to improve further as examiners gain more experience in their professional practice. This study suggests that the O-RADS US system could be used as an educational model in gynecologic residency programs, providing a foundation for training in other sonographic feature systems.

## Figures and Tables

**Figure 1 cancers-16-03820-f001:**
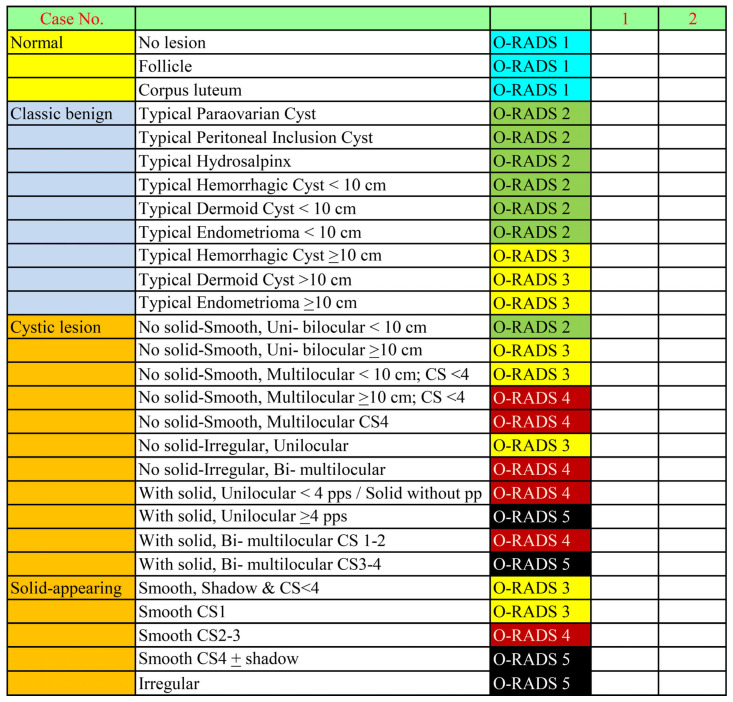
O-RADS checklist.

**Figure 2 cancers-16-03820-f002:**
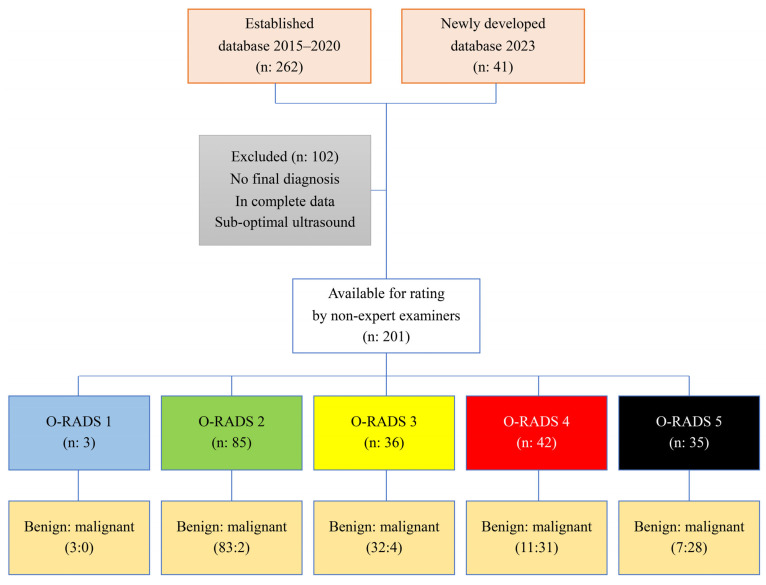
Flowchart of patient recruitment for O-RADS rating.

**Figure 3 cancers-16-03820-f003:**
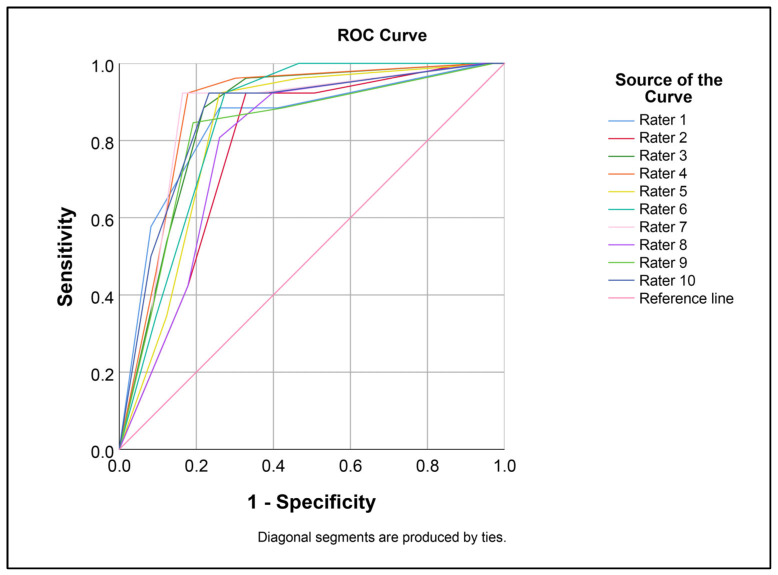
ROC curves demonstrating the performance in predicting malignant mass among ten raters The area under the curve values are not significantly different (Z-test, paired samples; all *p*-values > 0.05).

**Table 1 cancers-16-03820-t001:** Frequencies of various adnexal masses included in analysis.

Final Diagnosis	Number of Cases	Percentage
** *Benign Group* **		
Endometrioma	44	21.9
Mature cystic teratoma	27	13.4
Serous cystadenoma	16	8.0
Mucinous cystadenoma	11	5.5
Hemorrhagic cyst	8	4.0
Pseudocyst	8	4.0
Hydrosalpinx/tubo-ovarian abscess	7	3.5
Fibroma	5	2.5
Functional cyst	5	2.5
Simple epithelial cy	3	1.5
Brenner tumor	2	1.0
**Total**	**136**	**67.7**
** *Malignant Group* **		
Serous adenocarcinoma	15	7.5
Endometrioid adenocarcinoma	14	7.0
Mucinous adenocarcinoma	11	5.5
Mucinous low malignant potential	5	2.5
Dysgerminoma	3	1.5
Clear cell carcinoma	5	2.5
Yolk sac tumor	3	1.5
Immature teratoma	2	1.0
Sexcord stromal tumor	2	1.0
Metastatic carcinoma	1	0.5
Serous low malignant potential	1	0.5
Other cancers	3	1.5
**Total**	**65**	**32.3**

**Table 2 cancers-16-03820-t002:** Diagnostic performance of O-RADS in predicting malignant adnexal masses by each of ten raters and the consensus rater.

Rater	Area Under Curves	Sensitivity (95% CI)	Specificity (95% CI)
Rater 1	0.826 (0.747–0.905)	90.8 (82.2–96.2)	78.7 (71.3–85.0)
Rater 2	0.815 (0.737–0.893)	90.6 (81.9–96.2)	75.0 (67.3–81.8)
Rater 3	0.817 (0.735–0.899)	89.2 (80.2–95.2)	80.9 (73.7–86.9)
Rater 4	0.846 (0.767–0.925)	89.2 (80.2–95.2)	87.5 (81.3–92.3)
Rater 5	0.808 (0.723–0.894)	87.7 (77.6–94.5)	82.9 (75.6–88.9)
Rater 6	0.789 (0.702–0.876)	79.7 (68.8–88.3)	77.9 (70.5–84.3)
Rater 7	0.829 (0.743–0.915)	82.7 (71.0–91.3)	89.0 (82.5–93.8)
Rater 8	0.763 (0.668–0.858)	74.5 (62.1–84.8)	81.7 (74.4–87.8)
Rater 9	0.805 (0.717–0.892)	79.3 (67.7–88.3)	84.2 (77.4–89.7)
Rater 10	0.835 (0.757–0.913)	87.7 (78.3–94.2)	80.1 (72.9–86.3)
**Consensus Rater**	**0.841 (0.761–0.921)**	**90.8 (82.2–96.2)**	**86.8 (80.4–91.8)**

**Table 3 cancers-16-03820-t003:** Paired comparisons of areas under curve (AUC) values in predicting malignant mass among ten raters, expressed as mean difference in AUC.

The Paired Raters	AUC Difference (95% CI)	*p*-Value	The Paired Raters	AUC Difference (95% CI)	*p*-Value
Rater 1–Rater 2	0.011 (−0.050–0.071)	0.728	Rater 3–Rater 10	−0.018 (−0.081–0.044)	0.563
Rater 1–Rater 3	0.009 (−0.060–0.078)	0.806	Rater 4–Rater 5	0.038 (−0.031–0.107)	0.283
Rater 1–Rater 4	−0.020 (−0.091–0.050)	0.567	Rater 4–Rater 6	0.057 (−0.017–0.131)	0.128
Rater 1–Rater 5	0.017 (−0.069–0.104)	0.695	Rater 4–Rater 7	0.017 (−0.030–0.065)	0.477
Rater 1–Rater 6	0.037 (−0.040–0.113)	0.348	Rater 4–Rater 8	0.083 (−0.009–0.175)	0.077
Rater 1–Rater 7	−0.003 (−0.082–0.075)	0.937	Rater 4–Rater 9	0.042 (−0.041–0.124)	0.325
Rater 1–Rater 8	0.063 (−0.023–0.149)	0.152	Rater 4–Rater 10	0.011 (−0.035–0.056)	0.641
Rater 1–Rater 9	0.021 (−0.058–0.101)	0.603	Rater 5–Rater 6	0.019 (−0.061–0.100)	0.637
Rater 1–Rater 10	−0.010 (−0.077–0.058)	0.777	Rater 5–Rater 7	−0.020 (−0.091–0.050)	0.567
Rater 2–Rater 3	−0.002 (−0.053–0.049)	0.936	Rater 5–Rater 8	0.045 (−0.040–0.131)	0.298
Rater 2–Rater 4	−0.031 (−0.087–0.024)	0.269	Rater 5–Rater 9	0.004 (−0.070–0.077)	0.919
Rater 2–Rater 5	0.007 (−0.065–0.078)	0.857	Rater 5–Rater 10	−0.027 (−0.103–0.049)	0.483
Rater 2–Rater 6	0.026 (−0.050–0.102)	0.500	Rater 6–Rater 7	−0.040 (−0.116–0.036)	0.303
Rater 2–Rater 7	−0.014 (−0.079–0.051)	0.675	Rater 6–Rater 8	0.026 (−0.046–0.098)	0.478
Rater 2–Rater 8	0.052 (−0.040–0.144)	0.269	Rater 6–Rater 9	−0.016 (−0.098–0.067)	0.711
Rater 2–Rater 9	0.010 (−0.064–0.085)	0.784	Rater 6–Rater 10	−0.046 (−0.117–0.024)	0.197
Rater 2–Rater 10	−0.020 (−0.071–0.030)	0.429	Rater 7–Rater 8	0.066 (−0.028–0.160)	0.171
Rater 3–Rater 4	−0.029 (−0.092–0.034)	0.364	Rater 7–Rater 9	0.024 (−0.068–0.117)	0.607
Rater 3–Rater 5	0.009 (−0.062–0.079)	0.809	Rater 7–Rater 10	−0.007 (−0.066–0.053)	0.828
Rater 3–Rater 6	0.028 (−0.047–0.103)	0.465	Rater 8–Rater 9	−0.042 (−0.119–0.036)	0.292
Rater 3–Rater 7	−0.012 (−0.075–0.052)	0.715	Rater 8–Rater 10	−0.072 (−0.159–0.014)	0.102
Rater 3–Rater 8	0.054 (−0.029–0.137)	0.203	Rater 9–Rater10	−0.031 (−0.114–0.052)	0.466
Rater 3–Rater 9	0.012 (−0.068–0.093)	0.762			

**Table 4 cancers-16-03820-t004:** Diagnostic performance of O-RADS in predicting malignant adnexal masses by each of ten raters and the consensus rater.

	*p*-Value	Fleiss Kappa Index (95% CI)
**Agreement in rating malignancy**		
Overall rating	<0.001	0.642 (0.641–0.643)
■ Rating for benignity	<0.001	0.642 (0.641–0.643)
■ Rating for malignancy	<0.001	0.642 (0.641–0.643)
**Agreement in rating O-RADS category**		
Overall rating	<0.001	0.471 (0.470–0.471)
■ Rating for O-RADS 1	<0.001	0.416 (0.415–0.417)
■ Rating for O-RADS 2	<0.001	0.561 (0.561–0.562)
■ Rating for O-RADS 3	<0.001	0.357 (0.356–0.358)
■ Rating for O-RADS 4	<0.001	0.371 (0.370–0.372)
■ Rating for O-RADS 5	<0.001	0.526 (0.526–0.527)

**Table 5 cancers-16-03820-t005:** List of false-positive and false-negative diagnoses.

False-Positive Cases	Number	False-Negative Cases	Number
Complex endometrioma	6	Dermoid cyst with immature teratoma	2
Complicated dermoid cyst	4	Dermoid cyst with struma ovarii	1
Fibrotic tubo-ovarian abscess	3	Mucinous adeno CA	1
Mucinous cystadenoma	2	Sexcord stromal tumor	1
Serous cystadenoma	1	Endometriod carcinoma	1
Hemorrhagic mass	1		
Fibroma	1		
**Total**	**18**	**Total**	**6**

## Data Availability

The datasets analyzed during the current study are available from the corresponding author upon reasonable request.

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
