# Peer review of "Accuracy of O-RADS System in Differentiating Between Benign and Malignant Adnexal Masses Assessed via External Validation by Inexperienced Gynecologists"

_cancers, 2024, doi:10.3390/cancers16223820_

Round 1
Reviewer 1 Report
Comments and Suggestions for Authors
In general, the article is technically well written with a concise expression. The ultrasound assessment of adnexal masses is of great interest for obstetricians and health care providers, as ovarian cancer is an important cause of morbidity and mortality worldwide.
Regarding the research design, methods and results, I have a few questions and recommendations:
1. In the manuscript you say that “This study was conducted using a prospective database of gynecologic patients undergoing ultrasound examinations for adnexal masses … between March 2018 and August 2024.” but after that you said that “The study was ethically approved by the institutional review board” on 13 June 2023. Please try to clarify this, because it appears that you started a prospective study before you had the approval of the institutional review board.
2. In the manuscript you say that “These records were reviewed and displayed by the senior author (TT) to the ten inexperienced examiners (residents) for evaluation and O-RADS scoring.” – It is not clear if each record/case/patient was reviewed by one all of the residents. I understand you had 201 cases/patients, so one case was evaluated by one resident or one case was evaluated by all ten residents, meaning you had (201x10=) 2010 interpretations?
3. From what I understand you consider your study to be prospective? But you actually reviewed a database of images/video clips from an earlier study? I’m not sure if you can say that such a study is prospective, especially considering that your study was ethically approved on 13 June 2023 and you use data obtained prior to this date.
4. I did not understand who/what is the “consensus rater” and how did you get the results for him?
I support the acceptance of the manuscript after these revisions.
Author Response
Reviewer #1 (response with highlighted in red)
Comments and Suggestions for Authors
In general, the article is technically well written with a concise expression. The ultrasound assessment of adnexal masses is of great interest for obstetricians and health care providers, as ovarian cancer is an important cause of morbidity and mortality worldwide.
Regarding the research design, methods and results, I have a few questions and recommendations:
- In the manuscript you say that “This study was conducted using a prospective database of gynecologic patients undergoing ultrasound examinations for adnexal masses … between March 2018 and August 2024.” but after that you said that “The study was ethically approved by the institutional review board” on 13 June 2023. Please try to clarify this, because it appears that you started a prospective study before you had the approval of the institutional review board.
Response: Our department developed a prospective database of adnexal masses, including video clips, from 2018 to 2023 as an educational resource for ultrasound training. This study, which began in 2023, partially utilized video clips from the established database. The training course commenced in July 2023, with an evaluation of the trainees' O-RADS skills in July-August 2024. In revised MS we clarify this matter in the second paragraph of “Method”, as highlighted.
- In the manuscript you say that “These records were reviewed and displayed by the senior author (TT) to the ten inexperienced examiners (residents) for evaluation and O-RADS scoring.” – It is not clear if each record/case/patient was reviewed by one all of the residents. I understand you had 201 cases/patients, so one case was evaluated by one resident or one case was evaluated by all ten residents, meaning you had (201x10=) 2010 interpretations?
Response: A total of 201 cases (video clips) were displayed case by case to 10 residents simultaneously (approximately 8-10 cases per day for nearly a month; July-August 2024). This resulted in a total of about 2,000 interpretations for analysis. In revised MS we clarify this matter in the fifth paragraph of “Method”, as highlighted.
- From what I understand you consider your study to be prospective? But you actually reviewed a database of images/video clips from an earlier study? I’m not sure if you can say that such a study is prospective, especially considering that your study was ethically approved on 13 June 2023 and you use data obtained prior to this date.
Response: The main resources for the training course were video clips retrieved from the established prospective database. However, the training and assessments of these clips were conducted prospectively, and all 2,000 interpretations occurred afterward (July-August 2024). In the revised manuscript, we have clarified this issue more clearly, as highlighted.
- I did not understand who/what is the “consensus rater” and how did you get the results for him?
Response: Each case was assessed by 10 raters (residents), and the individual scoring results could vary. We used the median score from the 10 ratings to represent the collective evaluation and defined this as the consensus O-RADS score for each case. In revised MS we clarify this matter in “Method” subheading “Statistical analysis”, as highlighted.

Reviewer 2 Report
Comments and Suggestions for Authors
I have the following questions about this paper:
1. Despite the IOTA simple rules' high sensitivity and specificity, a considerable percentage of results (20–30%) were inconclusive. The author should discuss the implications of this and investigate how clinical decision-making and patient outcomes may be impacted by these inconclusive cases.
2. The author ought to investigate the reasons behind the disparity between the moderate agreement in O-RADS classification (Fleiss Kappa coefficient of 0.471) and the good agreement in predicting benign and malignant cases (Fleiss Kappa coefficient of 0.642).
3. The author may also discuss whether the lower level of agreement was caused by variations in image interpretation, training disparities, or the complexity of the O-RADS classification system. How to increase rater consistency in O-RADS categorization?
4. Since most diagnostic problems in clinical settings are encountered by less experienced practitioners, like general practitioners or general gynecologists, the author should discuss the significance of assessing the efficacy of predictive systems like IOTA simple rules, RMI, ADNEX models, or O-RADS US.
5. To ensure the wider applicability and dependability of these models in routine practice, the author may also think about addressing whether customized training programs or streamlined systems could improve diagnostic accuracy in these less experienced hands.
6. The theoretical advantage of including additional established clinical predictive factors in decision-making models, like menopausal status or CA-125 levels, should be further discussed by the author.
7. The author may offer a more thorough grasp of how multi-modal approaches might improve clinical outcomes by investigating whether the integration of these factors alongside sonographic features could improve diagnostic accuracy and patient management, especially in situations where sonographic assessment alone may be inconclusive.
Author Response
Reviewer #2 (response with highlighted in blue)
Comments and Suggestions for Authors
I have the following questions about this paper:
- Despite the IOTA simple rules' high sensitivity and specificity, a considerable percentage of results (20–30%) were inconclusive. The author should discuss the implications of this and investigate how clinical decision-making and patient outcomes may be impacted by these inconclusive cases.
Response: We have now included a brief comment on this matter in the 'Introduction,' as highlighted. However, it should be noted that the IOTA simple rules fall outside the scope of this study, which is focused on the O-RADS system. “Specifically, in cases where the IOTA simple rules yield inconclusive results, expert consultation or additional imaging modalities, such as MRI or CT, may be required.”
- The author ought to investigate the reasons behind the disparity between the moderate agreement in O-RADS classification (Fleiss Kappa coefficient of 0.471) and the good agreement in predicting benign and malignant cases (Fleiss Kappa coefficient of 0.642).
Response: We comment on the disparity in paragraph 5 “Discussion”, as highlighted, as follows.
- The author may also discuss whether the lower level of agreement was caused by variations in image interpretation, training disparities, or the complexity of the O-RADS classification system. How to increase rater consistency in O-RADS categorization?
Response: We discuss more on the disparity and how to improve the rater consistency in paragraph 5 “Discussion”, as suggested, as highlighted.
- Since most diagnostic problems in clinical settings are encountered by less experienced practitioners, like general practitioners or general gynecologists, the author should discuss the significance of assessing the efficacy of predictive systems like IOTA simple rules, RMI, ADNEX models, or O-RADS US.
Response: Other system is beyond scope of this study. However, we discuss on those system in “Introduction”.
- To ensure the wider applicability and dependability of these models in routine practice, the author may also think about addressing whether customized training programs or streamlined systems could improve diagnostic accuracy in these less experienced hands.
Response: Thank you for suggestion. We add the comment on this matter at the end of discussion (before conclusion), as suggested, as highlighted.
- The theoretical advantage of including additional established clinical predictive factors in decision-making models, like menopausal status or CA-125 levels, should be further discussed by the author.
Response: We have included a brief comment on the added value of such clinical findings in the 'Discussion' section under the subheading 'Clinical Implication,' as highlighted. Although the extent of this added value is not yet fully understood, we propose incorporating clinical factors into the O-RADS system as a future direction, as indicated in the 'Discussion' section under the subheading 'Research Implication,' as highlighted.
- The author may offer a more thorough grasp of how multi-modal approaches might improve clinical outcomes by investigating whether the integration of these factors alongside sonographic features could improve diagnostic accuracy and patient management, especially in situations where sonographic assessment alone may be inconclusive.
Response: We have included a brief comment on the added value of such clinical findings in the 'Discussion' section under the subheading 'Clinical Implication,' as highlighted. Although the extent of this added value is not yet fully understood, we propose incorporating clinical factors into the O-RADS system as a future direction, as indicated in the 'Discussion' section under the subheading 'Research Implication,' as highlighted.

Reviewer 3 Report
Comments and Suggestions for Authors
1. Firstly, the title of the article is too long and looks very tedious.
2. This article looks more like an experimental report than a research article. What is the significance and innovation of the research that the author should highlight in the article.
The English could be improved to more clearly express the research.
Author Response
1. Firstly, the title of the article is too long and looks very tedious. Response: The title is corrected as suggested by the reviewer 3 (in previous response) 2. This article looks more like an experimental report than a research article. What is the significance and innovation of the research that the author should highlight in the article. Response: The significance and innovation is summarized in the first paragraph of "Discussion" Finally, English has been checked by professional native speaker.Reviewer 4 Report
Comments and Suggestions for Authors
The authors have conducted a well-designed study.
I have only a few minor revisions.
1. Title: please change effectiveness to accuracy and shorten the length
2. Give 95%CI for sensitivity and specificity
3. Introduction:
1. Remove “.” after [2] and [3] in line 61
2. Line 77: Change ORADS and check over the text
4. Change effectiveness with accuracy
5. Did the authors evaluate to conduct a learning curve analysis to find out if and when each rater achieved a higher level of accuracy? It might be interesting to know.
6. Line 278: “ of” instead of “ or”.
Author Response
I have only a few minor revisions.
- Title: please change effectiveness to accuracy and shorten the length
Response: In revised MS, the title has been modified as suggested, as highlighted.
- Give 95%CI for sensitivity and specificity
Response: In revised MS, 95% CI for sensitivity and specificity are added as suggested, as highlighted.
- Introduction:
- Remove “.” after [2] and [3] in line 61
Response: In revised MS, “.” has been removed, as suggested.
- Line 77: Change ORADS and check over the text
Response: ORADS has been checked and changed to O-RADS throughout the manuscript, as suggested.
- Change effectiveness with accuracy
Response: In revised MS, “effectiveness” has been changed to “accuracy”, as suggested.
- Did the authors evaluate to conduct a learning curve analysis to find out if and when each rater achieved a higher level of accuracy? It might be interesting to know.
Response: We apologize that learning curve analysis could not be conducted in this study.
- Line 278: “ of” instead of “ or”.
Response: The word is now corrected. Thank you very much.
Round 2
Reviewer 3 Report
Comments and Suggestions for Authors
accept
Comments on the Quality of English Languagegood